# Neural Causal Models for Counterfactual Identification and Estimation

**Kevin Xia** and **Yushu Pan** and **Elias Bareinboim**
Causal Artificial Intelligence Laboratory
Columbia University, USA
{kevinmxia,yushupan,eb}@cs.columbia.edu

## Abstract

Evaluating hypothetical statements about how the world would be had a different course of action been taken is arguably one key capability expected from modern AI systems. Counterfactual reasoning underpins discussions in fairness, the determination of blame and responsibility, credit assignment, and regret. In this paper, we study the evaluation of counterfactual statements through neural models. Specifically, we tackle two causal problems required to make such evaluations, i.e., counterfactual identification and estimation from an arbitrary combination of observational and experimental data. First, we show that neural causal models (NCMs) are expressive enough and encode the structural constraints necessary for performing counterfactual reasoning. Second, we develop an algorithm for simultaneously identifying and estimating counterfactual distributions. We show that this algorithm is sound and complete for deciding counterfactual identification in general settings. Third, considering the practical implications of these results, we introduce a new strategy for modeling NCMs using generative adversarial networks. Simulations corroborate with the proposed methodology.

## 1 Introduction

Counterfactual reasoning is one of human's high-level cognitive capabilities, used across a wide range of affairs, including determining how objects interact, assigning responsibility, credit and blame, and articulating explanations. Counterfactual statements underpin prototypical questions of the form "what if–" and "why–", which inquire about hypothetical worlds that have not necessarily been realized (Pearl & Mackenzie, 2018). If a patient Alice had taken a drug and died, one may wonder, "why did Alice die?"; "was it the drug that killed her?"; "would she be alive had she not taken the drug?". In the context of fairness, why did an applicant, Joe, not get the job offer? Would the outcome have changed had Joe been a Ph.D.? Or perhaps of a different race? These are examples of fundamental questions about *attribution* and *explanation*, which evoke hypothetical scenarios that disagree with the current reality and which not even experimental studies can reconstruct.

We build on the semantics of counterfactuals based on a generative process called *structural causal model* (SCM) (Pearl, 2000). A fully instantiated SCM $\mathcal{M}^*$ describes a collection of causal mechanisms and distribution over exogenous conditions. Each $\mathcal{M}^*$ induces families of qualitatively different distributions related to the activities of seeing (called observational), doing (interventional), and imagining (counterfactual), which together are known as the *ladder of causation* (Pearl & Mackenzie, 2018; Bareinboim et al., 2022); also called the *Pearl Causal Hierarchy* (PCH). The PCH is a containment hierarchy in which distributions can be put in increasingly refined layers: observational content goes into layer 1 ($\mathcal{L}_1$); experimental to layer 2 ($\mathcal{L}_2$); counterfactual to layer 3 ($\mathcal{L}_3$). It is understood that there are questions about layers 2 and 3 that cannot be answered (i.e. are underdetermined), even given all information in the world about layer 1; further, layer 3 questions are still underdetermined given data from layers 1 and 2 (Bareinboim et al., 2022; Ibeling & Icard, 2020).

Counterfactuals represent the more detailed, finest type of knowledge encoded in the PCH, so naturally, having the ability to evaluate counterfactual distributions is an attractive proposition. In practice, a fully specified model $\mathcal{M}^*$ is almost never observable, which leads to the question – how can a counterfactual statement, from $\mathcal{L}_3^*$, be evaluated using a combination of observational and experimental data (from $\mathcal{L}_1^*$ and $\mathcal{L}_2^*$)? This question embodies the challenge of cross-layer inferences, which entail solving two challenging causal problems in tandem, identification and estimation.

In the more traditional literature of causal inference, there are different symbolic methods for solving these problems in various settings and under different assumptions. In the context of identification, there exists an arsenal of results that includes celebrated methods such as Pearl's do-calculus (Pearl, 1995), and go through different algorithmic methods when considering inferences for $\mathcal{L}_2$- (Tian & Pearl, 2002; Shpitser & Pearl, 2006; Huang & Valtorta, 2006; Bareinboim & Pearl, 2012; Lee et al., 2019; Lee & Bareinboim, 2020; 2021) and $\mathcal{L}_3$-distributions (Heckman, 1992; Pearl, 2001; Avin et al., 2005; Shpitser & Pearl, 2009; Shpitser & Sherman, 2018; Zhang & Bareinboim, 2018; Correa et al., 2021). On the estimation side, there are various methods including the celebrated Propensity Score/IPW for the backdoor case (Rubin,

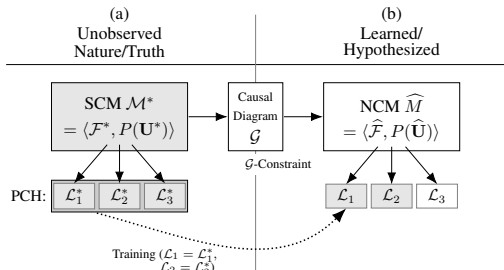

Figure 1: The l.h.s. contains the true SCM $\mathcal{M}^*$ that induces PCH's three layers. The r.h.s. contains a neural model $\widehat{M}$ constrained by inductive bias $\mathcal{G}$ (entailed by $\mathcal{M}^*$) and matching $\mathcal{M}^*$ on $\mathcal{L}_1$ and $\mathcal{L}_2$ through training.

1978; Horvitz & Thompson, 1952; Kennedy, 2019; Kallus & Uehara, 2020), and some more relaxed settings (Fulcher et al., 2019; Jung et al., 2020; 2021), but the literature is somewhat scarcer and less developed. In fact, there is a lack of estimation methods for $\mathcal{L}_3$ quantities in most settings.

On another thread in the literature, deep learning methods have achieved outstanding empirical success in solving a wide range of tasks in fields such as computer vision (Krizhevsky et al., 2012), speech recognition (Graves & Jaitly, 2014), and game playing (Mnih et al., 2013). One key feature of deep learning is its ability to allow inferences to scale with the data to high dimensional settings. We study here the suitability of the neural approach to tackle the problems of causal identification and estimation while trying to leverage the benefits of these new advances experienced in non-causal settings. [1] The idea behind the approach pursued here is illustrated in Fig. 1. Specifically, we will search for a neural model $\widehat{M}$ (r.h.s.) that has the same generative capability of the true, unobserved SCM $\mathcal{M}^*$ (l.h.s.); in other words, $\widehat{M}$ should be able to generate the same observed/inputted data, i.e., $\mathcal{L}_1 = \mathcal{L}_1^*$ and $\mathcal{L}_2 = \mathcal{L}_2^*$. [2] To tackle this task in practice, we use an inductive bias for the neural model in the form of a *causal diagram* (Pearl, 2000; Spirtes et al., 2000; Bareinboim & Pearl, 2016), which is a parsimonious description of the mechanisms ($\mathcal{F}^*$) and exogenous conditions ($P(\mathbf{U}^*)$) of the generating SCM. [3] The question then becomes: under what conditions can a model trained using this combination of qualitative inductive bias and the available data be suitable to answer questions about hypothetical counterfactual worlds, *as if* we had access to the true $\mathcal{M}^*$?

There exists a growing literature that leverages modern neural methods to solve causal inference tasks.[1] Our approach based on proxy causal models will answer causal queries by direct evaluation through a parameterized neural model $\widehat{M}$ fitted on the data generated by $\mathcal{M}^*$. [4] For instance, some recent work solves the estimation of interventional ($\mathcal{L}_2$) or counterfactual ($\mathcal{L}_3$) distributions from observational ($\mathcal{L}_1$) data in Markovian settings, implemented through architectures such as GANs, flows, GNNs, and VGAEs (Kocaoglu et al., 2018; Pawlowski et al., 2020; Zecevic et al., 2021; Sanchez-Martin et al., 2021). In some real-world settings, Markovianity is a too stringent condition (see discussion in App. D.4) and may be violated, which leads to the separation between layers 1 and 2, and, in turn, issues of causal identification. [5] The proxy approach discussed above was pursued in Xia et al. (2021) to solve the identification and estimation of interventional distributions ($\mathcal{L}_2$) from observational data ($\mathcal{L}_1$) in non-Markovian settings. [6] This work introduced an object we leverage throughout this paper called *Neural Causal Model* (NCM, for short), which is a class of SCMs constrained to neural network functions and fixed distributions over the exogenous variables. While

---

[1] One of our motivations is that these methods showed great promise at estimating effects from observational data under backdoor/ignorability conditions (Shalit et al., 2017; Louizos et al., 2017; Li & Fu, 2017; Johansson et al., 2016; Yao et al., 2018; Yoon et al., 2018; Kallus, 2020; Shi et al., 2019; Du et al., 2020; Guo et al., 2020).

[2] This represents an extreme case where all $\mathcal{L}_1$- and $\mathcal{L}_2$-distributions are provided as data. In practice, this may be unrealistic, and our method takes as input any arbitrary subset of distributions from $\mathcal{L}_1$ and $\mathcal{L}_2$.

[3] When imposed on neural models, they enforce equality constraints connecting layer 1 and layer 2 quantities, defined formally through the *causal Bayesian network* (CBN) data structure (Bareinboim et al., 2022, Def. 16).

[4] In general, $\widehat{M}$ does not need to, and will not be equal to the true SCM $\mathcal{M}^*$.

[5] Layer 3 differs from lower layers even in Markovian models; see Bareinboim et al. (2022, Ex. 7).

[6] Witty et al. (2021) shows a related approach taking the Bayesian route; further details, see Appendix C.

NCMs have been shown to be able to solve the identification and estimation tasks for $\mathcal{L}_2$ queries, their potential for counterfactual inferences is still largely unexplored, and existing implementations have been constrained to low-dimensional settings.

Despite all the progress achieved so far, no practical methods exist for estimating counterfactual ($\mathcal{L}_3$) distributions in the general setting where an arbitrary combination of observational ($\mathcal{L}_1$) and experimental ($\mathcal{L}_2$) distributions is available, and unobserved confounders exist (i.e. Markovianity does not hold). Hence, in addition to providing the first neural method of counterfactual identification, this paper establishes the first general counterfactual estimation technique even among non-neural methods, leveraging the neural toolkit for scalable inferences. Specifically, our contributions are:
1. We prove that when fitted with a graphical inductive bias, the NCM encodes the $\mathcal{L}_3$-constraints necessary for performing counterfactual inference (Thm. 1), and that they are still expressive enough to model the underlying data-generating model, which is not necessarily a neural network (Thm. 2).
2. We show that counterfactual identification within a neural proxy model setting is equivalent to established symbolic approaches (Thm. 3). We leverage this duality to develop an optimization procedure (Alg. 1) for counterfactual identification and estimation that is both sound and complete (Corol. 2). The approach is general in that it accepts any combination of inputs from $\mathcal{L}_1$ and $\mathcal{L}_2$, it works in any causal diagram setting, and it does not require the Markovianity assumption to hold.
3. We develop a new approach to modeling the NCM using generative adversarial networks (GANs) (Goodfellow et al., 2014), capable of robustly scaling inferences to high dimensions (Alg. 3). We then show how GAN-NCMs can solve the challenging optimization problems in identifying and estimating counterfactuals in practice. Experiments are provided in Sec. 5 and proofs in Appendix A. All supplementary material can be found in the full technical report (Xia et al., 2022).

**Preliminaries.** We now introduce the notation and definitions used throughout the paper. We use uppercase letters ($X$) to denote random variables and lowercase letters ($x$) to denote corresponding values. Similarly, bold uppercase ($\mathbf{X}$) and lower case ($\mathbf{x}$) letters are used to denote sets of random variables and values respectively. We use $\mathcal{D}_X$ to denote the domain of $X$ and $\mathcal{D}_{\mathbf{X}} = \mathcal{D}_{X_1} \times \cdots \times \mathcal{D}_{X_k}$ for the domain of $\mathbf{X} = \{X_1, \ldots, X_k\}$. We denote $P(\mathbf{X} = \mathbf{x})$ (which we will often shorten to $P(\mathbf{x})$) as the probability of $\mathbf{X}$ taking the values $\mathbf{x}$ under the probability distribution $P(\mathbf{X})$.

We utilize the basic semantic framework of structural causal models (SCMs), as defined in (Pearl, 2000, Ch. 7). An SCM $\mathcal{M}$ consists of endogenous variables $\mathbf{V}$, exogenous variables $\mathbf{U}$ with distribution $P(\mathbf{U})$, and mechanisms $\mathcal{F}$. $\mathcal{F}$ contains a function $f_{V_i}$ for each variable $V_i$ that maps endogenous parents $\mathbf{Pa}_{V_i}$ and exogenous parents $\mathbf{U}_{V_i}$ to $V_i$. Each $\mathcal{M}$ induces a causal diagram $\mathcal{G}$, where every $V_i \in \mathbf{V}$ is a vertex, there is a directed arrow $(V_j \rightarrow V_i)$ for every $V_i \in \mathbf{V}$ and $V_j \in \mathbf{Pa}_{V_i}$, and there is a dashed-bidirected arrow $(V_j \dashleftarrow\dashrightarrow V_i)$ for every pair $V_i, V_j \in \mathbf{V}$ such that $\mathbf{U}_{V_i}$ and $\mathbf{U}_{V_j}$ are not independent. For further details, see (Bareinboim et al., 2022, Def. 13/16, Thm. 4). The exogenous $\mathbf{U}_{V_i}$'s are not assumed independent (i.e. Markovianity is not required). Our treatment is constrained to *recursive* SCMs (implying acyclic causal diagrams) with finite domains over $\mathbf{V}$ (see Apps. A/E for details). Each SCM $\mathcal{M}$ assigns values to each counterfactual distribution as follows:

**Definition 1** (Layer 3 Valuation). An SCM $\mathcal{M}$ induces layer $\mathcal{L}_3(\mathcal{M})$, a set of distributions over $\mathbf{V}$, each with the form $P(\mathbf{Y}_*) = P(\mathbf{Y}_{1[\mathbf{x}_1]}, \mathbf{Y}_{2[\mathbf{x}_2], \ldots})$ such that

$$P^{\mathcal{M}}(\mathbf{y}_{1[\mathbf{x}_1]}, \mathbf{y}_{2[\mathbf{x}_2]}, \ldots) = \int_{\mathcal{D}_{\mathbf{U}}} \mathbb{1}\left[\mathbf{Y}_{1[\mathbf{x}_1]}(\mathbf{u}) = \mathbf{y}_1, \mathbf{Y}_{2[\mathbf{x}_2]}(\mathbf{u}) = \mathbf{y}_2, \ldots\right] dP(\mathbf{u}), \qquad (1)$$

where $\mathbf{Y}_{i[\mathbf{x}_i]}(\mathbf{u})$ is evaluated under $\mathcal{F}_{\mathbf{x}_i} := \{f_{V_j} : V_j \in \mathbf{V} \setminus \mathbf{X}_i\} \cup \{f_X \leftarrow x : X \in \mathbf{X}_i\}$. ∎

Each $\mathbf{Y}_i$ corresponds to a set of variables in a world where the original mechanisms $f_X$ are replaced with constants $\mathbf{x}_i$ for each $X \in \mathbf{X}_i$; this is also known as the mutilation procedure. This procedure corresponds to interventions, and we use subscripts to denote the intervening variables (e.g. $\mathbf{Y}_{\mathbf{x}}$) or subscripts with brackets when the variables are indexed (e.g. $\mathbf{Y}_{1[\mathbf{x}_1]}$). For instance, $P(y_x, y'_{x'})$ is the probability of the joint counterfactual event $Y = y$ had $X$ been $x$ and $Y = y'$ had $X$ been $x'$.

SCM $\mathcal{M}_2$ is said to be $P^{(\mathcal{L}_i)}$-consistent (for short, $\mathcal{L}_i$-consistent) with SCM $\mathcal{M}_1$ if $\mathcal{L}_i(\mathcal{M}_1) = \mathcal{L}_i(\mathcal{M}_2)$. We will use $\mathbb{Z}$ to denote a set of quantities from Layer 2 (i.e. $\mathbb{Z} = \{P(\mathbf{V}_{\mathbf{z}_k})\}_{k=1}^{\ell}$), and we use $\mathbb{Z}(\mathcal{M})$ to denote those same quantities induced by SCM $\mathcal{M}$ (i.e. $\mathbb{Z}(\mathcal{M}) = \{P^{\mathcal{M}}(\mathbf{V}_{\mathbf{z}_k})\}_{k=1}^{\ell}$).

We use neural causal models (NCMs) as a substitute (proxy) model for the true SCM, as follows:

**Definition 2** ($\mathcal{G}$-Constrained Neural Causal Model ($\mathcal{G}$-NCM) (Xia et al., 2021, Def. 7)). Given a causal diagram $\mathcal{G}$, a $\mathcal{G}$-constrained Neural Causal Model (for short, $\mathcal{G}$-NCM) $\widehat{M}(\boldsymbol{\theta})$ over variables $\mathbf{V}$

with parameters $\boldsymbol{\theta} = \{\theta_{V_i} : V_i \in \mathbf{V}\}$ is an SCM $\langle \widehat{\mathbf{U}}, \mathbf{V}, \widehat{\mathcal{F}}, \widehat{P}(\widehat{\mathbf{U}})\rangle$ such that $\widehat{\mathbf{U}} = \{\widehat{U}_\mathbf{C} : \mathbf{C} \in \mathbb{C}(\mathcal{G})\}$, where $\mathbb{C}(\mathcal{G})$ is the set of all maximal cliques over bidirected edges of $\mathcal{G}$, and $\mathcal{D}_{\widehat{U}} = [0,1]$ for all $\widehat{U} \in \widehat{\mathbf{U}}$; $\widehat{\mathcal{F}} = \{\hat{f}_{V_i} : V_i \in \mathbf{V}\}$, where each $\hat{f}_{V_i}$ is a feedforward neural network parameterized by $\theta_{V_i} \in \boldsymbol{\theta}$ mapping values of $\mathbf{U}_{V_i} \cup \mathbf{Pa}_{V_i}$ to values of $V_i$ for $\mathbf{U}_{V_i} = \{\widehat{U}_\mathbf{C} : \widehat{U}_\mathbf{C} \in \widehat{\mathbf{U}} \text{ s.t. } V_i \in \mathbf{C}\}$ and $\mathbf{Pa}_{V_i} = Pa_\mathcal{G}(V_i)$; $\widehat{P}(\widehat{\mathbf{U}})$ is defined s.t. $\widehat{U} \sim \text{Unif}(0,1)$ for each $\widehat{U} \in \widehat{\mathbf{U}}$. ∎

## 2 NEURAL CAUSAL MODELS FOR COUNTERFACTUAL INFERENCE

We first recall that inferences about higher layers of the PCH generated by the true SCM $\mathcal{M}^*$ cannot be made in general through an NCM $\widehat{M}$ trained only from lower layer data (Bareinboim et al., 2022; Xia et al., 2021). This impossibility motivated the use of the inductive bias in the form of a causal diagram $\mathcal{G}$ in the construction of the NCM in Def. 2, which ascertains that the $\mathcal{G}$-consistency property holds. (See App. D.1 for further discussion.) We next define consistency w.r.t. to each layer, which will be key for a more fine-grained discussion later on.

**Definition 3** ($\mathcal{G}^{(\mathcal{L}_i)}$-Consistency). Let $\mathcal{G}$ be the causal diagram induced by the SCM $\mathcal{M}^*$. For any SCM $\mathcal{M}$, $\mathcal{M}$ is said to be $\mathcal{G}^{(\mathcal{L}_i)}$-consistent (w.r.t. $\mathcal{M}^*$) if $\mathcal{L}_i(\mathcal{M})$ satisfies all layer $i$ equality constraints implied by $\mathcal{G}$. ∎

This generalization is subtle since regardless of which $\mathcal{L}_i$ is used with the definition, the causal diagram $\mathcal{G}$ generated by $\mathcal{M}^*$ is the same. The difference lies in the implied constraints. For instance, if an SCM $\mathcal{M}$ is $\mathcal{G}^{(\mathcal{L}_1)}$-consistent, that means that $\mathcal{G}$ is a Bayesian network for the observational distribution of $\mathcal{M}$, implying independences readable through d-separation Pearl (1988). If $\mathcal{M}$ is $\mathcal{G}^{(\mathcal{L}_2)}$-consistent, that means that $\mathcal{G}$ is a *Causal Bayesian network* (CBN) (Bareinboim et al., 2022, Def. 16) for the interventional distributions of $\mathcal{M}$. While several SCMs could share the same d-separation constraints as $\mathcal{M}^*$, there are fewer that share all $\mathcal{L}_2$ constraints encoded by the CBN. $\mathcal{G}$-consistency at higher layers imposes a stricter set of constraints, narrowing down the set of compatible SCMs. There also exist constraints of layer 3 that are important for counterfactual inferences.

To motivate the use of such constraints, consider an example inspired by the multi-armed bandit problem. A casino has 3 slot machines, labeled "0", "1", and "2". Every day, the casino assigns one machine a good payout, one a bad payout, and one an average payout, with chances of winning represented by exogenous variables $U_+$, $U_-$, and $U_=$, respectively. A customer comes every day and plays a slot machine. $X$ represents their choice of machine, and $Y$ is a binary variable representing whether they win. Suppose a data scientist creates a model of the situation, and she hypothesizes that the casino predicts the customer's choice based on their mood ($U_M$) and will always assign the predicted machine the average payout to maintain profits. Her model is described by the SCM $\mathcal{M}'$:

$$\mathcal{M}' = \begin{cases} \mathbf{U} & = \{U_M, U_+, U_=, U_-\}, U_M \in \{0,1,2\}, U_+, U_=, U_- \in \{0,1\} \\ \mathbf{V} & = \{X, Y\}, X \in \{0,1,2\}, Y \in \{0,1\} \\ \mathcal{F} & = \begin{cases} f_X(u_M) & = u_M \\ f_Y(x, u_M, u_+, u_=, u_-) & = \begin{cases} u_= & x = u_M \\ u_- & x = (u_M - 1)\%3 \\ u_+ & x = (u_M + 1)\%3 \end{cases} \end{cases} \\ P(\mathbf{U}): & P(U_M = i) = \frac{1}{3}, P(U_+ = 1) = 0.6, P(U_= = 1) = 0.4, P(U_- = 1) = 0.2 \end{cases} \tag{2}$$

It turns out that in this model $P(y_x) = P(y \mid x)$. For example, $P(Y = 1 \mid X = 0) = P(U_= = 1) = 0.4$, and $P(Y_{X=0} = 1) = P(U_M = 0)P(U_= = 1) + P(U_M = 1)P(U_- = 1) + P(U_M = 2)P(U_+ = 1) = \frac{1}{3}(0.4) + \frac{1}{3}(0.2) + \frac{1}{3}(0.6) = 0.4$.

Suppose the true model $\mathcal{M}^*$ employed by the casino (and unknown by the customers and data scientist) induces graph $\mathcal{G} = \{X \rightarrow Y\}$. Interestingly enough, $\mathcal{M}'$ would be $\mathcal{G}^{(\mathcal{L}_2)}$-consistent with $\mathcal{M}^*$ since $\mathcal{M}'$ is compatible with all $\mathcal{L}_2$-constraints, including $P(y_x) = P(y \mid x)$ and $P(x_y) = P(x)$. However, and perhaps surprisingly, it would fail to be $\mathcal{G}^{(L_3)}$-consistent. A further constraint implied by $\mathcal{G}$ on the third layer is that $P(y_x \mid x') = P(y_x)$, which is not true of $\mathcal{M}'$. To witness, note that $P(Y_{X=0} = 1 \mid X = 2) = P(U_+ = 1) = 0.6$ in $\mathcal{M}'$, which means that if the customer chose machine 2, they would have had higher payout had they chosen machine 0. This does not match $P(Y_{X=0} = 1) = 0.4$, computed earlier, so $\mathcal{M}'$ fails to encode the $\mathcal{L}_3$-constraints implied by $\mathcal{G}$.

In general, the causal diagram encodes a family of $\mathcal{L}_3$-constraints which we leverage to make cross-layer inferences. A more detailed discussion can be found in Appendix D. We show next that NCMs encodes all of the equality constraints related to $\mathcal{L}_3$, in addition to the known $\mathcal{L}_2$-constraints.

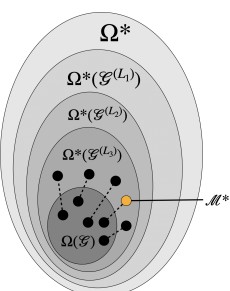

**Theorem 1** (NCM $\mathcal{G}^{(\mathcal{L}_3)}$-Consistency). *Any $\mathcal{G}$-NCM $\widehat{M}(\boldsymbol{\theta})$ is $\mathcal{G}^{(\mathcal{L}_3)}$-consistent.* ∎

This will be a key result for performing inferences at the counterfactual level. Similar to how constraints about layer 2 distributions help bridge the gap between layers 1 and 2, layer 3 constraints allow us to extend our inference capabilities into layer 3. (In fact, most of $\mathcal{L}_3$'s distributions are not obtainable through experimentation.) While this graphical inductive bias is powerful, the set of NCMs constrained by $\mathcal{G}$ is no less expressive than the set of SCMs constrained by $\mathcal{G}$, as shown next.

Figure 2: Model-theoretic visualization of Thms. 1 and 2.

**Theorem 2** ($\mathcal{L}_3$-$\mathcal{G}$ Expressiveness). *For any SCM $\mathcal{M}^*$ that induces causal diagram $\mathcal{G}$, there exists a $\mathcal{G}$-NCM $\widehat{M}(\boldsymbol{\theta}) = \langle \widehat{\mathbf{U}}, \mathbf{V}, \widehat{\mathcal{F}}, \widehat{P}(\widehat{\mathbf{U}}) \rangle$ s.t. $\widehat{M}$ is $\mathcal{L}_3$-consistent w.r.t. $\mathcal{M}^*$.* ∎

This result ascertains that the NCM class is as expressive, and therefore, contains the same generative capabilities as the original generating model. More interestingly, even if the original SCM $\mathcal{M}^*$ does not belong to the NCM class, but from the higher space, there exists a NCM $\widehat{M}(\boldsymbol{\theta})$ that will be capable of expressing the collection of distributions from all layers of the PCH induced by it.

A visual representation of these two results is shown in Fig. 2. The space of all SCMs is called $\Omega^*$, and the subspace that contains all SCMs $\mathcal{G}^{((\mathcal{L}_i)}$-consistent w.r.t. the true SCM $\mathcal{M}^*$ (black dot) is called $\Omega^*(\mathcal{G}^{(\mathcal{L}_i)})$. Note that the $\mathcal{G}^{(\mathcal{L}_i)}$ space shrinks with higher layers, indicating a more constrained space with fewer SCMs. Thm. 1 states that all $\mathcal{G}$-NCMs ($\Omega(\mathcal{G})$) are within $\Omega^*(\mathcal{G}^{(\mathcal{L}_3)})$, and Thm. 2 states that all SCMs in $\Omega^*(\mathcal{G}^{(\mathcal{L}_3)})$ can be represented by a corresponding $\mathcal{G}$-NCM on all three layers.

It may seem intuitive that the $\mathcal{G}$-NCM has these two properties by construction, but these properties are nontrivial and, in fact, not enjoyed by many model classes. Examples can be found in Appendix D. Together, these two theorems ensure that the NCM has both the constraints and the expressiveness necessary for counterfactual inference, elaborated further in the next section.

## 3 NEURAL COUNTERFACTUAL IDENTIFICATION

The problem of identification is concerned with determining whether a certain quantity is computable from a combination of assumptions, usually encoded in the form of a causal diagram, and a collection of distributions (Pearl, 2000, p. 77). This challenge stems from the fact that even though the space of SCMs (or NCMs) is constrained upon assuming a certain causal diagram, the quantity of interest may still be underdetermined. In words, there are many SCMs compatible with the same diagram $\mathcal{G}$ but generate different answers for the target distribution. In this section, we investigate the problem of identification and decide whether counterfactual quantities (from $\mathcal{L}_3$) can be inferred from a combination of a subset of $\mathcal{L}_2$ and $\mathcal{L}_1$ datasets together with $\mathcal{G}$, as formally defined next.

**Definition 4** (Neural Counterfactual Identification). Consider an SCM $\mathcal{M}^*$ and the corresponding causal diagram $\mathcal{G}$. Let $\mathbb{Z} = \{P(\mathbf{V}_{\mathbf{z}_k})\}_{k=1}^{\ell}$ be a collection of available interventional (or observational if $\mathbf{Z}_k = \emptyset$) distributions from $\mathcal{M}^*$. The counterfactual query $P(\mathbf{Y}_* = \mathbf{y}_* \mid \mathbf{X}_* = \mathbf{x}_*)$ is said to be neural identifiable (identifiable, for short) from the set of $\mathcal{G}$-constrained NCMs $\Omega(\mathcal{G})$ and $\mathbb{Z}$ if and only if $P^{\widehat{M}_1}(\mathbf{y}_* \mid \mathbf{x}_*) = P^{\widehat{M}_2}(\mathbf{y}_* \mid \mathbf{x}_*)$ for every pair of models $\widehat{M}_1, \widehat{M}_2 \in \Omega(\mathcal{G})$ s.t. they match $\mathcal{M}^*$ on all distributions in $\mathbb{Z}$ (i.e. $\mathbb{Z}(\mathcal{M}^*) = \mathbb{Z}(\mathcal{M}_1) = \mathbb{Z}(\mathcal{M}_2) > 0$). ∎

From a symbolic standpoint, a counterfactual quantity $P(\mathbf{y}_* \mid \mathbf{x}_*)$ is identifiable from $\mathcal{G}$ and $\mathbb{Z}$ if all SCMs that induce the distributions of $\mathbb{Z}$ and abide by the constraints of $\mathcal{G}$ also agree on $P(\mathbf{y}_* \mid \mathbf{x}_*)$. This is illustrated in Fig. 3. In the definition above, the search is constrained to the NCM subspace (shown in light gray) within the space of SCMs (dark gray). It may be concerning that the true SCM $\mathcal{M}^*$ might not be an NCM, as we alluded to earlier. The next result ascertains that identification within the constrained space of NCMs is actually equivalent to identification in the original SCM-space.

**Theorem 3** (Counterfactual Graphical-Neural Equivalence (Dual ID)). *Let $\Omega^*, \Omega$ be the spaces including all SCMs and NCMs, respectively. Consider the true SCM $\mathcal{M}^*$ and the corresponding*

*causal diagram $\mathcal{G}$. Let $Q = P(\mathbf{y}_* \mid \mathbf{x}_*)$ be the target query and $\mathbb{Z}$ the set of observational and interventional distributions available. Then, $Q$ is neural identifiable from $\Omega(\mathcal{G})$ and $\mathbb{Z}$ if and only if it is identifiable from $\mathcal{G}$ and $\mathbb{Z}$.* ∎

Interestingly, this result connects the new concept of neural counterfactual identification (Def. 4) with established non-neural results. If a counterfactual quantity is determined to be neural identifiable, then it is also identifiable from $\mathcal{G}$ and $\mathbb{Z}$ through other non-neural means, and vice versa.[7] Practically speaking, counterfactual inference can be performed while constrained in the NCM space, and the obtained results will be faithful to existing symbolic approaches. This broadens the previous results connecting NCMs to classical identification.

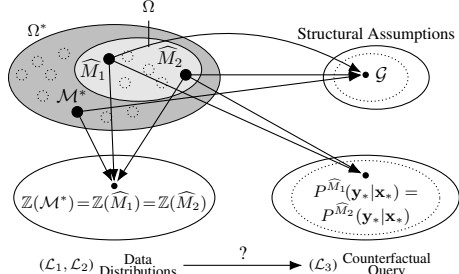

**Corollary 1** (Neural Counterfactual Mutilation (Operational ID)). *Consider the true SCM $\mathcal{M}^* \in \Omega^*$, causal diagram $\mathcal{G}$, a set of available distributions $\mathbb{Z}$, and a target query $Q$ equal to $P^{\mathcal{M}^*}(\mathbf{y}_* \mid \mathbf{x}_*)$. Let $\widehat{M} \in \Omega(\mathcal{G})$ be a $\mathcal{G}$-constrained NCM such that $\mathbb{Z}(\widehat{M}) = \mathbb{Z}(\mathcal{M}^*)$. If $Q$ is identifiable from $\mathcal{G}$ and $\mathbb{Z}$, then $Q$ is computable via Eq. 1 from $\widehat{M}$.* ∎

Figure 3: $P(\mathbf{y}_*)$ is identifiable from $\mathbb{Z}$ and $\Omega(\mathcal{G})$ if for any SCM $\mathcal{M}^* \in \Omega^*$ and NCMs $\widehat{M}_1, \widehat{M}_2 \in \Omega$ (top left), $\widehat{M}_1, \widehat{M}_2, \mathcal{M}^*$ match in $\mathbb{Z}$ (bottom left) and $\mathcal{G}$ (top right), then the NCMs $\widehat{M}_1, \widehat{M}_2$ also match in $P(\mathbf{y}_*)$ (bottom right).

Corol. 1 states that once identification is established, the counterfactual query can be inferred through the NCM $\widehat{M}$, as if it were the true SCM $\mathcal{M}^*$, by directly applying layer 3's definition to $\widehat{M}$ (Def. 1). Remarkably, this result holds even if $\mathcal{M}^*$ does not match $\widehat{M}$ in either the mechanisms $\mathcal{F}$ or the exogenous dist. $P(\mathbf{U})$, and it only requires some specific properties: $\mathcal{G}^{(\mathcal{L}_3)}$-consistency, matching $\mathbb{Z}$, and identifiability. Without these properties, inferences performed on $\widehat{M}$ would bear no meaningful information about the ground truth. To understand this subtlety, refer to examples in App. D.

Building on these results, we demonstrate through the procedure **NeuralID** (Alg. 1) how to decide the identifiability of counterfactual quantities. The specific optimization procedure searches explicitly in the space of NCMs for two models that respectively minimize and maximize the target query while maintaining consistency with the provided data distributions in $\mathbb{Z}$. If the two models match in the target query $Q$, then the effect is identifiable, and the value is returned; otherwise, the effect is non-identifiable.

---

**Algorithm 1**: **NeuralID** – Identifying/estimating counterfactual queries with NCMs.

**Input** : query $Q = P(\mathbf{y}_*|\mathbf{x}_*)$, $\mathcal{L}_2$ datasets $\mathbb{Z}(\mathcal{M}^*)$, and causal diagram $\mathcal{G}$

**Output** : $P^{\mathcal{M}^*}(\mathbf{y}_*|\mathbf{x}_*)$ if identifiable, FAIL otherwise.

1   $\widehat{M} \leftarrow \text{NCM}(\mathbf{V}, \mathcal{G})$      // from Def. 2
2   $\boldsymbol{\theta}_{\min}^* \leftarrow \arg\min_{\boldsymbol{\theta}} P^{\widehat{M}(\boldsymbol{\theta})}(\mathbf{y}_*|\mathbf{x}_*)$ s.t. $\mathbb{Z}(\widehat{M}(\boldsymbol{\theta})) = \mathbb{Z}(\mathcal{M}^*)$
3   $\boldsymbol{\theta}_{\max}^* \leftarrow \arg\max_{\boldsymbol{\theta}} P^{\widehat{M}(\boldsymbol{\theta})}(\mathbf{y}_*|\mathbf{x}_*)$ s.t. $\mathbb{Z}(\widehat{M}(\boldsymbol{\theta})) = \mathbb{Z}(\mathcal{M}^*)$
4   **if** $P^{\widehat{M}(\boldsymbol{\theta}_{\min}^*)}(\mathbf{y}_*|\mathbf{x}_*) \neq P^{\widehat{M}(\boldsymbol{\theta}_{\max}^*)}(\mathbf{y}_*|\mathbf{x}_*)$ **then**
5      **return** FAIL
6   **else**
7      **return** $P^{\widehat{M}(\boldsymbol{\theta}_{\min}^*)}(\mathbf{y}_*|\mathbf{x}_*)$    // choose min or max arbitrarily

---

The implementation of how to enforce these consistency constraints in practice is somewhat challenging. We note two nontrivial details that are abstracted away in the description of Alg. 1. First, although training to fit a single, observational dataset is straightforward, it is not as clear how to simultaneously maintain consistency with the multiple datasets in $\mathbb{Z}$. Second, unlike with simpler interventional queries, it is not clear how to search the parameter space in a way that maximizes or minimizes a counterfactual query, which may be more involved due to nesting (e.g. $P(Y_{Z_{X=0}})$) or evaluating the same variable in multiple worlds (e.g. $P(Y_{X=0}, Y_{X=1})$). The details of how to solve these issues are discussed in Sec. 4.

Interestingly, this approach is qualitatively different than the case of classical, symbolic methods that avoid operating in the space of SCMs directly. Still, in principle, this alternative approach does not imply any loss in functionality, as evident from the next result.

**Corollary 2** (Soundness and Completeness). *Let $\Omega^*$ be the set of all SCMs, $\mathcal{M}^* \in \Omega^*$ be the true SCM inducing causal diagram $\mathcal{G}$, $Q = P(\mathbf{y}_* \mid \mathbf{x}_*)$ be a query of interest, and $\widehat{Q}$ be the result from*

---

[7]We say identification from $\mathcal{G}$ and $\mathbb{Z}$ instead of $\Omega^*(\mathcal{G})$ and $\mathbb{Z}$ because existing symbolic approaches (e.g. do-calculus) directly solve the identification problem on top of the graph instead of the space of SCMs.

*running Alg. 1 with inputs $\mathbb{Z}(\mathcal{M}^*) > 0$, $\mathcal{G}$, and $Q$. Then $Q$ is identifiable from $\mathcal{G}$ and $\mathbb{Z}$ if and only if $\widehat{Q}$ is not FAIL. Moreover, if $\widehat{Q}$ is not FAIL, then $\widehat{Q} = P^{\mathcal{M}^*}(\mathbf{y}_* \mid \mathbf{x}_*)$.* ∎

In words, the procedure **NeuralID** is both necessary and sufficient in this very general setting, implying that for any instances involving any arbitrary diagram, datasets, or queries, the identification status of the query is always classified correctly by this algorithm.

## 4 NEURAL COUNTERFACTUAL ESTIMATION

We developed a procedure for identifying counterfactual quantities that is both sound and complete under ideal conditions – e.g., unlimited data, perfect optimization, which is encouraging. In this section, we build on these results and establish a more practical approach to solving these tasks under imperfect optimization and finite samples.

---

**Algorithm 2**: NCM Counterfactual Sampling

**Input** : NCM
$\widehat{M}(\boldsymbol{\theta}) = \langle \widehat{\mathbf{U}}, \mathbf{V}, \widehat{\mathcal{F}}, P(\widehat{\mathbf{U}}) \rangle$,
counterfactual $\mathbf{Y}_*$, conditional
$\mathbf{X}_* = \mathbf{x}_*$, number of samples $m$

**Output** : $m$ samples from $P^{\widehat{M}}(\mathbf{Y}_* | \mathbf{x}_*)$

1   **Function** $\widehat{M}$.sample($\mathbf{Y}_*, \mathbf{x}_*, m$):
2     $S \leftarrow \emptyset$
3     **while** $|S| < m$ **do**
4       $\widehat{\mathbf{u}} \leftarrow P(\widehat{\mathbf{U}})$.sample()
5       **if** $\mathbf{X}_*^{\widehat{M}(\boldsymbol{\theta})}(\widehat{\mathbf{u}}) = \mathbf{x}_*$ **then**
6         $S$.add($\mathbf{Y}_*^{\widehat{M}(\boldsymbol{\theta})}(\widehat{\mathbf{u}})$)
7     **return** $S$;

---

Consider a $\mathcal{G}$-NCM $\widehat{M}$ constructed as specified by Def. 2. Any counterfactual statement $\mathbf{Y}_* = (\mathbf{Y}_{1[\mathbf{x}_1]}, \mathbf{Y}_{2[\mathbf{x}_2]}, \dots)$ can be evaluated from an NCM $\widehat{M}$ for a specific setting of $\widehat{\mathbf{U}} = \widehat{\mathbf{u}}$ by computing the corresponding values of $\mathbf{Y}_i$ in the mutilated submodel $\widehat{M}_{\mathbf{x}}$ for each $i$. That is,

$$\mathbf{Y}_*^{\widehat{M}}(\mathbf{u}) = (\mathbf{Y}_{1[\mathbf{x}_1]}^{\widehat{M}}(\mathbf{u}), \mathbf{Y}_{2[\mathbf{x}_2]}^{\widehat{M}}(\mathbf{u}), \dots) \qquad (3)$$

Then, sampling can be done following the natural approach delineated by Alg. 2. In words, the distribution $P(\mathbf{Y}_* \mid \mathbf{x}_*)$ can be sampled from NCM $\widehat{M}$ by (1) sampling instances of $P(\widehat{\mathbf{U}})$, (2) computing their corresponding value for $\mathbf{X}_*$ (via Eq. 3) while rejecting cases that do not match $\mathbf{x}_*$, and (3) returning the corresponding value for $\mathbf{Y}_*$ (via Eq. 3) for the remaining instances.

Following this procedure, a counterfactual $P(\mathbf{Y}_* = \mathbf{y}_* \mid \mathbf{X}_* = \mathbf{x}_*)$ can be estimated from the NCM through a Monte-Carlo approach, instantiating Eq. 4 as follows:

$$P^{\widehat{M}}(\mathbf{y}_* \mid \mathbf{x}_*) \approx \frac{\sum_{j=1}^m \mathbb{1}\left[\mathbf{Y}_*^{\widehat{M}}(\widehat{\mathbf{u}}_j) = \mathbf{y}_*, \mathbf{X}_*^{\widehat{M}}(\widehat{\mathbf{u}}_j) = \mathbf{x}_*\right]}{\sum_{j=1}^m \mathbb{1}\left[\mathbf{X}_*^{\widehat{M}}(\widehat{\mathbf{u}}_j) = \mathbf{x}_*\right]}, \qquad (4)$$

where $\{\widehat{\mathbf{u}}_j\}_{j=1}^m$ are a set of $m$ samples from $P(\widehat{\mathbf{U}})$.

Alg. 3 demonstrates how to solve the challenging optimization task in lines 2 and 3 of Alg. 1. The first step is to learn parameters such that the distributions induced by the NCM $\widehat{M}$ match the true distributions in $\mathbb{Z}$. While Alg. 1 describes the inputs in the form of $\mathcal{L}_2$-distributions, $\mathbb{Z}(\mathcal{M}^*) = \{P^{\mathcal{M}^*}(\mathbf{V}_{\mathbf{z}_k})\}_{k=1}^\ell$, in most settings, one has the empirical versions of such distributions in the form of finite datasets, $\{\widehat{P}^{\mathcal{M}^*}(\mathbf{V}_{\mathbf{z}_k}) = \{\mathbf{v}_{\mathbf{z}_k,i}\}_{i=1}^{n_k}\}_{k=1}^\ell$.

One way to train $\widehat{M}$ to match $\mathcal{M}^*$ in the distribution $P(\mathbf{V}_{\mathbf{z}_k})$ is to compare the distribution of data points in $\widehat{P}^{\mathcal{M}^*}(\mathbf{V}_{\mathbf{z}_k})$ with the distribution of samples from $\widehat{M}$, $\widehat{P}^{\widehat{M}}(\mathbf{V}_{\mathbf{z}_k})$. The two empirical distributions can be compared using a divergence function $\mathbb{D}_P$, which returns a smaller value when the two distributions are "similar". The goal is then to minimize $\mathbb{D}_P(\widehat{P}^{\widehat{M}}(\mathbf{V}_{\mathbf{z}_k}), \widehat{P}^{\mathcal{M}^*}(\mathbf{V}_{\mathbf{z}_k}))$ for each $k \in \{1, \dots, \ell\}$. In this work, a generative adversarial approach (Goodfellow et al., 2014) is taken to train the NCM, and $\mathbb{D}_P$ is computed using a discriminator network.

In addition to fitting the datasets, the second challenge of Alg. 1 is to simultaneously maximize or minimize the query of interest $Q = P(\mathbf{y}_* \mid \mathbf{x}_*)$. This can be done by first computing samples of $P(\mathbf{Y}_* \mid \mathbf{x}_*)$ from $\widehat{M}$ via Alg. 2, denoted $\widehat{Q}$, and then minimizing (or maximizing) the "distance" between $Q$ and $\widehat{Q}$. Essentially, samples from $\mathbf{Y}_*$ are penalized based on how similar they are to the correct values $\mathbf{y}_*$. For example, if the query to maximize is $P(Y = 1)$, and a value of 0.6 is sampled for $Y$ from $\widehat{M}$, then the goal could be to minimize squared error, $(1 - 0.6)^2$. In general, a distance metric $\mathbb{D}_Q$ is used to compute the distance between $\widehat{Q}$ and $Q$, and we use log loss for $\mathbb{D}_Q$ as our experiments involve binary variables.

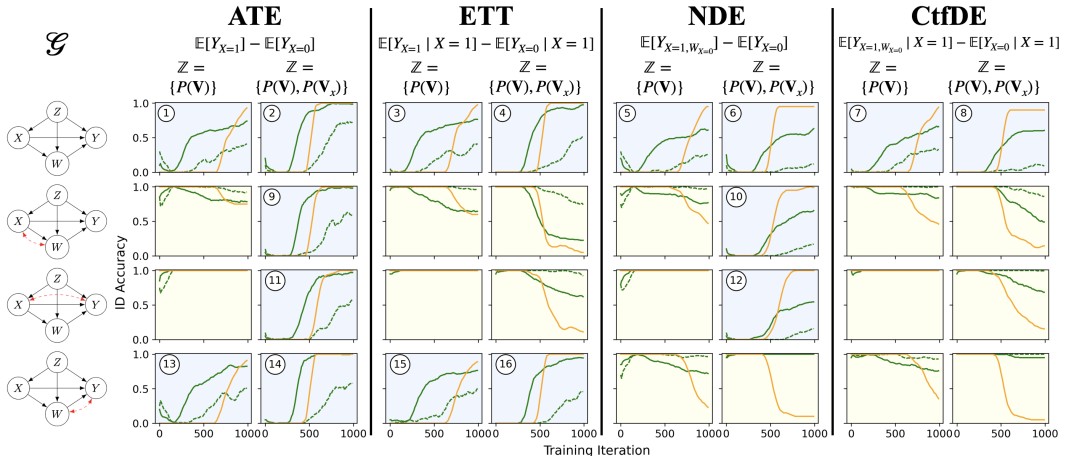

Figure 4: Experimental results on deciding identifiability on counterfactual queries with NCMs. GAN-NCM (green) is compared with MLE-NCM (orange) for settings with $d = 1$. GAN-NCM performance is also shown for $d = 16$ (dashed green). Blue (resp. yellow) backgrounds on plots correspond to a ground truth of ID (resp. non-ID). ID cases are numbered for reference in later plots.

For this reason, an NCM trained with this approach will be referred to as a *GAN-NCM*. [8] More details about architecture and hyperparameters used throughout this work can be found in Appendix B.

Putting $\mathbb{D}_P$ and $\mathbb{D}_Q$ together, we can write that the objective $L\left(\widehat{M}, \{\widehat{P}^{\mathcal{M}^*}(\mathbf{V}_{\mathbf{z}_k})\}_{k=1}^{\ell}\right)$ is

$$\left(\sum_{k=1}^{\ell} \mathbb{D}_P\left(\widehat{P}_{\mathbf{z}_k}^{\widehat{M}}, \widehat{P}_{\mathbf{z}_k}^{\mathcal{M}^*}\right)\right) \pm \lambda \mathbb{D}_Q\left(\widehat{Q}, Q\right), \tag{5}$$

where $\lambda$ is initially set to a high value, and decreases during training. Optimization may be done using gradient descent. After training, the two values of $Q$ induced by $\widehat{M}(\boldsymbol{\theta}_{\min})$ and $\widehat{M}(\boldsymbol{\theta}_{\max})$ are compared with a hypothesis testing procedure to decide identifiability. Eq. 4 is used as $Q$'s estimate, whenever identifiable.

---

**Algorithm 3**: Training Model

**Input** : Data $\{\widehat{P}^{\mathcal{M}^*}(\mathbf{V}_{\mathbf{z}_k}) = \{\mathbf{v}_{\mathbf{z}_k,i}\}_{i=1}^{n_k}\}_{k=1}^{\ell}$, query $Q = P(\mathbf{y}_*|\mathbf{x}_*)$, causal diagram $\mathcal{G}$, number of Monte Carlo samples $m$, regularization constant $\lambda$, learning rate $\eta$, training epochs $T$

1  $\widehat{M} \leftarrow \text{NCM}(\mathbf{V}, \mathcal{G})$        // from Def. 2
2  Initialize parameters $\boldsymbol{\theta}_{\min}$ and $\boldsymbol{\theta}_{\max}$
3  **for** $t \leftarrow 1$ **to** $T$ **do**
4     $L_{\min} \leftarrow 0, L_{\max} \leftarrow 0$
5     **for** $k \leftarrow 1$ **to** $\ell$ **do**
        // Sample via Alg. 2
6       $\widehat{P}_{\min}(\mathbf{V}_{\mathbf{z}_k}) \leftarrow \widehat{M}(\boldsymbol{\theta}_{\min}).\text{sample}(\mathbf{V}_{\mathbf{z}_k}, n_k)$
7       $\widehat{P}_{\max}(\mathbf{V}_{\mathbf{z}_k}) \leftarrow \widehat{M}(\boldsymbol{\theta}_{\max}).\text{sample}(\mathbf{V}_{\mathbf{z}_k}, n_k)$
8       $L_{\min} \leftarrow L_{\min} + \mathbb{D}_P\left(\widehat{P}_{\min}(\mathbf{V}_{\mathbf{z}_k}), \widehat{P}^{\mathcal{M}^*}(\mathbf{V}_{\mathbf{z}_k})\right)$
9       $L_{\max} \leftarrow L_{\max} + \mathbb{D}_P\left(\widehat{P}_{\max}(\mathbf{V}_{\mathbf{z}_k}), \widehat{P}^{\mathcal{M}^*}(\mathbf{V}_{\mathbf{z}_k})\right)$
10    $\widehat{Q}_{\min} \leftarrow \widehat{M}(\boldsymbol{\theta}_{\min}).\text{sample}(\mathbf{Y}_*, m)$
11    $\widehat{Q}_{\max} \leftarrow \widehat{M}(\boldsymbol{\theta}_{\max}).\text{sample}(\mathbf{Y}_*, m)$
     // $L$ from Eq. 5
12    $L_{\min} \leftarrow L_{\min} - \lambda \mathbb{D}_Q\left(\widehat{Q}_{\min}, Q\right)$
13    $L_{\max} \leftarrow L_{\max} + \lambda \mathbb{D}_Q\left(\widehat{Q}_{\max}, Q\right)$
14    $\boldsymbol{\theta}_{\min} \leftarrow \boldsymbol{\theta}_{\min} - \eta \nabla L_{\min}$
15    $\boldsymbol{\theta}_{\max} \leftarrow \boldsymbol{\theta}_{\max} - \eta \nabla L_{\max}$

---

## 5 EXPERIMENTAL EVALUATION

We first evaluate the NCM's ability to identify counterfactual distributions through Alg. 3. [9] Each setting consists of a target query ($Q$), a causal diagram ($\mathcal{G}$), and a set of input distributions ($\mathbb{Z}$). In total, we test 32 variations. Specifically, we evaluate the identifiability of four queries $Q$: (1) Average Treatment Effect (ATE), (2) Effect of Treatment on the Treated (ETT) (Pearl, 2000, Eq. 8.18), (3) Natural Direct Effect (NDE) (Pearl, 2001, Eq. 6), and (4) Counterfactual Direct Effect (CtfDE) (Zhang & Bareinboim, 2018, Eq. 3); each expression is shown on the top of Fig. 4. The four graphs used are shown on the figure's left side, and represent general structures found throughout the mediation and fairness literature (Pearl, 2001; Zhang & Bareinboim, 2018). The variable $X$ encodes the treatment/decision, $Y$ the outcome, $Z$ observed features, and $W$ mediating variables. Lastly, we consider a setting in which only the observational data is available ($\mathbb{Z} = \{P(\mathbf{V})\}$) and another in which additional experimental data on $X$ is available ($\mathbb{Z} = \{P(\mathbf{V}), P(\mathbf{V}_x)\}$). In the experiments shown, all variables are 1-dimensional binary variables except $Z$, whose dimensionality $d$ is adjusted

---

[8]Other choices of $\mathbb{D}_P$ include KL divergence or Maximum Mean Discrepancy (MMD) (Gretton et al., 2012).
[9]The code is publicly available at: https://github.com/CausalAILab/NCMCounterfactuals

in experiments. The background color of each setting indicates that the query $Q$ is identifiable (blue) or is not identifiable (yellow) from the inputted $\mathcal{G}$ and $\mathbb{Z}$. Given the sheer volume of variations, we summarize the experiments below and provide further discussion and details in Appendix B.

We implement two approaches to ground the discussion around NCMs, one based on GANs (*GAN-NCM*), and another based on maximum likelihood (*MLE-NCM*). The former was discussed in the previous section and the latter is quite natural in statistical settings. The experiments (Fig. 4) show that GAN-NCM has on average higher accuracy. The MLE-NCM performs slightly better in ID cases (blue), but the performance drops significantly for non-ID cases (yellow), suggesting it may be biased in returning ID for all cases. The GAN-NCM is also shown to achieve decent performance in 16-d, where the MLE-NCM fails to work. We plot the run time of these two approaches in Fig. 5, which shows that the MLE-NCM scales poorly compared to the GAN-NCM; this pattern is observed in all settings. Intuitively, this is not surprising since the MLE-NCM explicitly computes a likelihood for every value in every variable domain, the size of which grows exponentially w.r.t. the dimensionality $(d)$, while the GAN-NCM avoids this by implicitly fitting distributions through $P(\widehat{\mathbf{U}})$ and $\widehat{\mathcal{F}}$ and directly outputting samples.

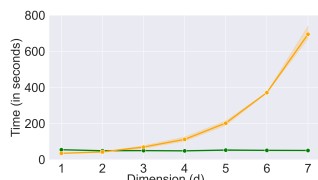

Figure 5: Results comparing run times of 100 epochs of training between GAN-NCM (green) and MLE-NCM (orange) in the first graph of Fig. 4 as the dimensionality $d$ of $Z$ scales higher.

For the identifiable cases (blue background), the target $Q$ is estimated through Eq. 4 after training. Results are shown in Fig. 6. The MLE-NCM serves as a benchmark for 1-dimensional cases since, intuitively, the data distributions can be learned more accurately when modeled explicitly. Still, even when $d = 1$, the GAN-NCM achieves competitive results in most settings and consistently achieves an error under 0.05 with more samples. The GAN-NCM is able to maintain this consistency even at $d = 16$, demonstrating its robustness scaling to higher dimensions.

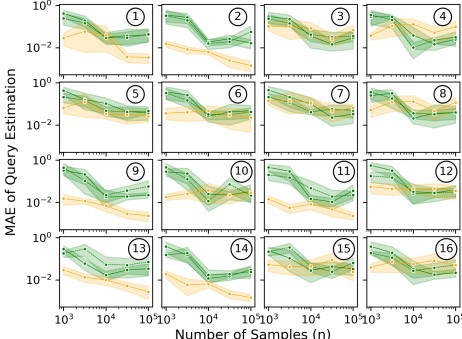

Figure 6: Results on estimating identifiable cases from Fig. 4 (corresponding numbers shown on the right). Mean Absolute Error (MAE) is plotted (with 95% confidence) for each setting for varying sample sizes. Results are shown for GAN-NCM (solid green) and MLE-NCM (orange) with $d = 1$ and also GAN-NCM with $d = 16$ (dashed green).

After all, the GAN-NCM is shown to be effective at identifying and estimating counterfactual distributions even in high dimensions. As expected, the MLE-NCM may achieve lower error in some 1-d settings, but the GAN-NCM may be preferred for scalability. Moreover, an incorrect ID conclusion in a non-ID case may be dangerous for downstream decision-making as the resulting estimation will likely be incorrect or misleading. The GAN-NCM is evidently more robust in such non-ID cases while still performing competitively in ID cases. Further experiments and discussions are provided in App. B.

## 6 CONCLUSIONS

We developed in this work a neural approach to the problems of counterfactual identification and estimation using neural causal models (NCMs). Specifically, we first showed that with the graphical inductive bias, NCMs are capable of encoding counterfactual ($\mathcal{L}_3$) constraints and are still expressive so as to represent any generating SCM (Thms. 1, 2). We then showed that NCMs have the ability of solving any counterfactual identification instance (Thm. 3, Corol 1). Given these theoretical properties, we introduced a sound and complete algorithm (Alg. 1, Corol. 2) for identifying and estimating counterfactuals in general non-Markovian settings given arbitrary datasets from $\mathcal{L}_1$ and $\mathcal{L}_2$. We developed an approach based on GANs to implement this algorithm in practice (Alg. 3) and empirically demonstrated its ability to scale inferences. From a neural perspective, counterfactual reasoning under a causal inductive bias allows for deep models to be trained with an improved understanding of interpretability and generalizability. From a causal perspective, neural nets can now provide tools to solve counterfactual inference problems previously only understood in theory.

ACKNOWLEDGEMENTS

This research was supported in part by the NSF, ONR, AFOSR, DoE, Amazon, JP Morgan, and The Alfred P. Sloan Foundation.

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
