# OpenReview forum: "Neural Causal Models for Counterfactual Identification and Estimation"
_ICLR.cc/2023/Conference — ICLR 2023 poster_

### Official Review · Reviewer_D1b5 · 2022-10-23

**Confidence:** 2
**Correctness:** 4
**Technical Novelty And Significance:** 4
**Empirical Novelty And Significance:** 2
**Recommendation:** 6

**Clarity, Quality, Novelty And Reproducibility:**

The authors have jammed tons of materials within the page limit and this has severely decreased readability of the paper. The results of the paper are novel for a conference publication. The experimental section is not extensive and reproducibility is not a key issue for this paper.

**Strength And Weaknesses:**

The overall arguments of the paper seem intuitive and correct. Although, the authors do not provide short overview of the proofs and the paper looks a bit jammed (better fits a journal paper). The paper heavily relies on the familiarity of the audience with the recent prior works such as (Bareinboim et al., 2022). Thus, some of the statements of the paper are unclear and need more explanations. E.g., "Our treatment is constrained to recursive SCMs, which implies acyclic causal diagrams, with finite domains over V."
The use of the term neural seems to be mostly for marketing purposes and the results of the paper may hold for general function estimators.


**Summary Of The Paper:**

This paper advances the theory of neural causal models and prove that they can recover the counterfactuals (L3 in Pearl's ladder of causality). The authors also provide new identification results and Algorithm 1 that uses supervised learning techniques to tell if the causal prior constraints are sufficient for identification. Finally, they propose GAN-NCM to scale causal models to high dimensional data.

**Summary Of The Review:**

This paper advances the theory of neural causal models and prove that they can recover the counterfactuals (L3 in Pearl's ladder of causality). The authors also provide new identification results and Algorithm 1 that uses supervised learning techniques to tell if the causal prior constraints are sufficient for identification. Finally, they propose GAN-NCM to scale causal models to high dimensional data.

---

> ### Author Response · Authors · 2022-11-11
> **Response to Reviewer D1b5**
>
> We thank the reviewer for taking time to provide valuable feedback and comments. We appreciate the generally positive reception of our work, and hope that the following responses will clarify some of your concerns.
>
> > Although, the authors do not provide short overview of the proofs and the paper looks a bit jammed (better fits a journal paper).
>
> > The authors have jammed tons of materials within the page limit and this has severely decreased readability of the paper.
>
> **Reply:** Indeed, there were many exciting aspects of our work that we unfortunately had to leave in the Appendix due to space constraints. This includes the proofs of the theoretical results in Appendix A (p. 15-18), which provide valuable insights to the discussed problems. We do have many results, and understand that the main body may be a bit short to allow readers to fully appreciate some aspects of our work. For that reason, we tried to add extensive material in Appendix D (p. 39-53) with several examples to support each result discussed in the main body. We have also added a FAQ in Appendix E (p. 54-57) to answer potential questions more quickly without requiring the reader to parse the entire supplementary material. If there is anything unclear about the work, we hope that the answer can be found in these appendices. Certainly, we would be happy to answer any further questions that the reviewer might have or consider re-prioritizing other points that may deserve to be promoted/demoted from/to the appendices.
>
> > The paper heavily relies on the familiarity of the audience with the recent prior works such as (Bareinboim et al., 2022). Thus, some of the statements of the paper are unclear and need more explanations. E.g., "Our treatment is constrained to recursive SCMs, which implies acyclic causal diagrams, with finite domains over V."
>
> **Reply:** Our work does indeed build on top of key results of other works, and we apologize for any confusion arising from such references. In order to keep the paper focused on the contributions of this work, as mentioned above, we deferred lengthy explanations of the other results to Appendix A (p. 15-18) and only included the aspects in the main body that were important to understanding the new results. In the particular case you mentioned, that statement is simply written there to be transparent about the assumptions, even though this assumption is somewhat standard in the literature and many times used even without being explicitly stated. These only come into play in the proofs in Appendix A, so we did not feel it was necessary to expand on that statement in the main body. If you feel the presentation is too deficient in this matter, one option is to add further elaboration and other pointers behind this assumption.
>
> > The use of the term neural seems to be mostly for marketing purposes and the results of the paper may hold for general function estimators.
>
> **Reply:** Many of the theoretical results of the paper do indeed hold for other architectures, not necessarily only neural ones, so long as they satisfy certain properties. We reference in Q4 of Appendix E that the specific properties are discussed in Xia, Lee, Bengio, Bareinboim (2021), which introduced the NCM data structure that we leverage in our paper. We focus on the “neural” aspect of the model to ground the discussion to a concrete model type, since it would be difficult to understand the impact of the work without a realizable model in practice. While other options are possible, neural networks are known to be effective in high-dimensional data settings, and Sec. 4 is dedicated to explaining how to implement and optimize such a model in practice, utilizing the neural toolkit. To be clear, we are not trying to decide for the user the best choice of model, and certainly they can rely on the results of this paper for other architectural choices and topologies.

---

### Official Review · Reviewer_452t · 2022-10-24

**Confidence:** 3
**Correctness:** 4
**Technical Novelty And Significance:** 4
**Empirical Novelty And Significance:** 2
**Recommendation:** 8

**Clarity, Quality, Novelty And Reproducibility:**

As far as I understand, the main contribution is on theory. I suggest the authors emphasize what the challenge is for their main result. For example, for me, Thm. 1 seems to be an evident result, because the neural causal model is just to use many feedforward neural networks to model each causal mechanism P(V_i|pa(V_i)) according to my understanding of Def. 2. Could the authors give some intuition about why it is not evident, or does I miss or misunderstand some important facts?

My other questions:
1. Could the authors explain "It may be concerning that the true SCM $M^*$ might not be an NCM, as we alluded to earlier." further? Where does it allude?
2. How to determine Line 4 of Alg. 1?
3. Under the current framework, does the proposed algorithm only apply when the data is discrete?

**Strength And Weaknesses:**

Strength:
1. Counterfactual inference is an interesting topic. This paper takes deep learning techniques into counterfactual inference, which could inspire more related studies.

2. The paper presents solid theoretical results.

Weaknesses:
The paper is hard to read. I understand that there are many new results, but it will be better if the authors could clearly illustrate the value of their theoretical results and the challenge.

**Summary Of The Paper:**

This paper builds the basic theoretical result for neural counterfactual inference. Based on the theoretical results, the authors present the algorithm to determine the identification of counterfacual and give the algorithm to estimate it if it is identifiable.

**Summary Of The Review:**

After the rebuttal, I increase my score.

#-----------
Currently, I can only give a baseline score because I am not sure about the studies in this paper. I look forward to the response of the authors.

---

> ### Author Response · Authors · 2022-11-11
> **Response to Reviewer 452t (1/2)**
>
> We thank the reviewer for the valuable feedback and interesting questions and comments. We also appreciate your willingness to engage in a thoughtful discussion and hope the following responses will clarify your concerns.
>
> > The paper is hard to read. I understand that there are many new results, but it will be better if the authors could clearly illustrate the value of their theoretical results and the challenge.
>
> > As far as I understand, the main contribution is on theory. I suggest the authors emphasize what the challenge is for their main result. For example, for me, Thm. 1 seems to be an evident result, because the neural causal model is just to use many feedforward neural networks to model each causal mechanism P(V_i|pa(V_i)) according to my understanding of Def. 2. Could the authors give some intuition about why it is not evident, or does I miss or misunderstand some important facts?
>
> **Reply:** There were many aspects of the work that we were excited to include but had to move to the Appendix due to space constraints. We thank you for the suggestion, and while we cannot fit more explanation in the main body, we will ensure that there are clear references to the Appendix to supplement any result in the main body whose contribution is unclear. If you have any specific suggestion on what to emphasize further, we would be happy to incorporate it as well.
>
> Going now to your question, the theory part is certainly a big contribution of our work (Secs. 2 and 3), but we also provide empirical contributions to demonstrate the theoretical results in practice (Secs. 4 and 5). This is outlined by the three bullets at the end of Sec. 1 above the preliminaries, where the first two bullets indicate theoretical contributions while the third indicates empirical contributions.
>
> Broadly speaking, the first bullet states that the $\mathcal{G}$-NCM model class has both the critical properties of constraint and expressiveness (Thms. 1 and 2) on the counterfactual level, which are not enjoyed by many model classes. In particular, regarding your specific question, Thm. 1 states that the NCM satisfies all constraints of $\mathcal{G}$, without which one cannot make any claims about higher layers due to the Causal Hierarchy Theorem. Many works that use the causal diagram (see Appendix C for examples) achieve this property but take it for granted as it is not discussed formally. In contrast, we formally discuss the implications of graphical constraints in Sec. 2 and Appendix A.1. Furthermore, the interesting aspect of the $\mathcal{G}$-NCM is that it simultaneously achieves both the properties of Thm. 1 and Thm. 2. Thm. 2 ascertains that within the constrained space, NCMs are still maximally expressive. This is important given that the true model could be any possible SCM, and the constraints (imposed by Thm. 1) do not imply lack of generality.
>
> Possibly interesting to our discussion, we would like to note that it is straightforward to come up with examples that satisfy one but not both properties. Example 4 in Appendix D.3 is an example of a model class that does not satisfy the property of Thm. 1, and this is due to the fact that a simple collection of neural networks, although expressive, lacks the property of parsimony. As discussed in the example, attempting to incorporate the proper constraints in a naive way will result in an intractable amount of optimization. Examples 5 and 6 are examples of model classes that do not satisfy the property of Thm. 2. It is only through the design of the NCM via Def. 2 that provides it the ability to simultaneously incorporate both of these properties in a natural and intuitive way, allowing it to solve the tasks of causal identification and estimation as discussed in later sections. Q5 of Appendix E summarizes these points.
>
> Finally, in response to your comment about the causal mechanisms corresponding to $P(V_i \mid pa(V_i))$, we note that since we allow for cases with unobserved confounders (no Markovianity assumption), the distribution $P(\mathbf{V})$ does not, in general, factorize into terms of $P(V_i \mid pa(V_i))$. So, the functions learned by the $\mathcal{G}$-NCM do not necessarily correspond to these conditional probabilities. In fact, there is no guarantee that they match anything w.r.t. the true model’s functions even after perfect training. Even so, Thm. 3 and Corol. 1 establish an important duality which states that the trained $\mathcal{G}$-NCM will output the correct value for the query, so long as it is identifiable. We expand on the reasons why the lack of Markovianity is particularly challenging for this problem in Appendix D.4. Certainly, we believe that solving the problem without the Markovianity assumption is nontrivial and worth studying, and is one of the strengths of the proposed work.

---

> > ### Author Response · Authors · 2022-11-11
> > **Response to Reviewer 452t (2/2)**
> >
> > > Could the authors explain "It may be concerning that the true SCM might not be an NCM, as we alluded to earlier." further? Where does it allude?
> >
> > **Reply:** This is indeed a subtle point and we appreciate the opportunity of providing further elaboration. That sentence alludes to Thm. 2 and its implications, discussed in the paragraph just below Thm. 2. The theorem states that even though $\mathcal{G}$-NCMs are forced to take specific parametric forms (e.g. uniform noise, neural net functions), they are still capable of expressing any arbitrary SCM that induces $\mathcal{G}$ on all three layers (and which are not necessarily NCMs).
> >
> > In the context of classical identification theory, if a counterfactual query is identifiable, that implies that all SCMs that match in the given data and graph will also match in the query. On the other hand, a query is ‘neural identifiable’ if all *NCMs* that match in the given data and graph also match in the query. Note here that the universal quantifier is over the space of NCMs, and not SCMs. This is possibly concerning since the NCM space is not the same as the SCM space, so perhaps a property (e.g., identifiability) may hold for NCMs but may not hold for the space of SCMs. Thm. 3 highlights the connection between these spaces and guarantees that because NCMs can fully express the space of SCMs (Thm. 2), there is no loss of generality when solving the identification problem within the space of NCMs, instead of SCMs. In fact, this is not the case for other model classes. For instance, consider a class of models restricted to linear functions. Perhaps all linear models that match the data and graph agree on a query, but there is an SCM with nonlinear functions that disagrees. This means that inferences within the linear class cannot be translated to the broader space that includes arbitrary SCMs. This is certainly not special to linear and also applies to many other classes (for instance, see Examples 5 and 6 in Appendix D.3). After all, there are subtle issues about the relationship across spaces, depending on the specific model class, and Thm. 3 guarantees that constraining the inferences to the NCM space is valid and imposes no loss even when the true model is not within it.
> >
> > > How to determine Line 4 of Alg. 1?
> >
> > **Reply:** Thanks for the question. In practice, the induced query values for the two models are unlikely to be exactly the same in identifiable cases (due to imperfect optimization, finite samples, limited training, etc.), so there is a need to incorporate a hypothesis testing procedure. The procedure is mentioned at the bottom of Sec. 4 and is explained in detail in Appendix B.4. In summary, the same trial is rerun several times, and the gaps between the queries of the max model and min model are collected. The result is considered to be identifiable if, with 95% confidence, the gap is under a threshold, $\tau$, which is set to $0.05$ in the experiments. The choice of $\tau$ is up to the practitioner, with lower values resulting in more false negatives and higher values resulting in more false positives.
> >
> > > Under the current framework, does the proposed algorithm only apply when the data is discrete?
> >
> > **Reply:** Indeed, the formal results of our work builds on other results that have only been proven for the discrete case. While similar results could potentially be proven for the continuous case, this would be a nontrivial extension that would require a different formalization from established results in counterfactual inference. For instance, we rely on discretization of SCM results, starting from (Balke and Pearl, 1994), and more recently (Zhang, Tian, Bareinboim, 2022), to prove Thm. 2.
> >
> > Empirically, however, if the query can be sampled via Alg. 2, then similar techniques applied to our paper can be extended without much change. Consider the example we provide in Q10 of Appendix E: if the query is $E[Y_{X=x} | X \geq 0]$ with both continuous $X$ and $Y$, we can (1) sample several instances of $X$, (2) filter out negative samples, (3) sample $Y$ using the values of $\mathbf{U}$ from the remaining instances, and (4) compute the average out of these values of $Y$. Then, this Monte Carlo estimate can be used in Alg. 3 for the purpose of maximizing/minimizing the query as well as estimating the final result if identifiable.
> >
> > Still, we have not made a full claim given the lack of proper foundations. We believe these are great directions for future research, and we would be interested in seeing how the NCM can be adapted to solve problems with continuous variables, in addition to trying to do this ourselves. We will reflect this note more explicitly in the updated manuscript.

---

> > > ### Comment · Reviewer_452t · 2022-11-12
> > > **Thank you for your response!**
> > >
> > > Dear authors: I have read every word of your rebuttal, and it indeed addresses my questions and helps me a lot. For me, this paper is exciting, impressive, and inspiring. And the methods and results are powerful. Thank you very much for your so detailed and clear response. I increase my score, and I am very happy to see it accepted!

---

> > > > ### Author Response · Authors · 2022-11-14
> > > > **RE: Thank you for your response!**
> > > >
> > > > We appreciate the questions and the opportunity of clarifying the raised points, thank you!

---

### Official Review · Reviewer_AWWR · 2022-10-26

**Confidence:** 2
**Clarity, Quality, Novelty And Reproducibility:** The paper is novel and the proposed s…
**Correctness:** 3
**Technical Novelty And Significance:** 3
**Empirical Novelty And Significance:** 3
**Recommendation:** 8

**Strength And Weaknesses:**

In general, the paper is well-written, and it aims to solve an important and interesting problem. The experiment can support the claims in the paper, and the authors have provided a very detailed appendix, which can help to understand their ideas.
I do not find significant weaknesses in this paper.

**Summary Of The Paper:**

Basically, this paper aims to study the problem of evaluating the counterfactual statements based on neural models. To achieve this goal, the authors firstly detail the previous work on the causal neural models, and then propose to identify counterfactual quantities based on the first two causal layers. The authors have also proved the soundness and completeness, which makes the paper very solid.


**Summary Of The Review:**

see the above comments

---

> ### Author Response · Authors · 2022-11-11
> **Response to Reviewer AWWR**
>
> We appreciate the reviewer for the positive assessment and encouraging comments. We were glad the problem was well-motivated and the work was understood. We hope that others will build upon this work and develop further interesting strategies for using neural networks for counterfactual inference.

---

### Decision · Program_Chairs · 2023-01-20

**Decision:**

Accept: poster

**Justification For Why Not Higher Score:**

Overlap with other existing work, like Xia et al., removes some of the impact (but there are more than enough new ideas here).

**Justification For Why Not Lower Score:**

It tackles a broad class of problems of central importance in causal inference while being nicely written and of practical relevance, and easy to use in principle.

**Metareview: Summary, Strengths And Weaknesses:**

The paper introduces new algorithms for estimating counterfactual quantities, such as as natural direct effects. In particular, identifiability is not necessary, and as such it extends the scope of causal inference to many problems of interest where knowledge provided is still informative even if it does not imply a unique answer.

Strengths: it tackles a broad class of problems of central importance in causal inference while being nicely written and of practical relevance, and easy to use in principle.

Weaknesses: overlap with other existing work, like Xia et al., removes some of the impact (but there are more than enough new ideas here). I would also recommend citing Hu, Y., Wu, Y., Zhang, L., and Wu, X. (2021). A generative adversarial framework for bounding confounded causal effects, AAAI, which covers some similar ground.

**Note From Pc:**

if the above contains the word "oral" or "spotlight" please see: "oral" presentation means -> notable-top-5% and "spotlight" means -> notable-top-25%. As stated in our emails, we are disassociating presentation type from AC recommendations